# Deriving time-concordant event cascades from gene expression data: A case study for Drug-Induced Liver Injury (DILI)

**Anika Liu**[1,2,3]*, **Namshik Han**[1,4], **Jordi Munoz-Muriedas**[2,5], **Andreas Bender**[3]*

**1** Milner Therapeutics Institute, University of Cambridge, Cambridge, United Kingdom, **2** Systems Modelling and Translational Biology, Data and Computational Sciences, GSK, London, United Kingdom, **3** Centre for Molecular Informatics, Department of Chemistry, University of Cambridge, Cambridge, United Kingdom, **4** Cambridge Centre for AI in Medicine, Department of Applied Mathematics and Theoretical Physics, University of Cambridge, Cambridge, United Kingdom, **5** Computer-Aided Drug Design, UCB, Slough, United Kingdom

* al862@cam.ac.uk (AL); ab454@cam.ac.uk (AB)

**Data Availability Statement:** Histopathology data was derived from the supplementary information of Sutherland et al. (2018), accessed through https://doi.org/10.1038/tpj.2017.17. TG-GATEs gene

## Abstract

Adverse event pathogenesis is often a complex process which compromises multiple events ranging from the molecular to the phenotypic level. In toxicology, Adverse Outcome Pathways (AOPs) aim to formalize this as temporal sequences of events, in which event relationships should be supported by causal evidence according to the tailored Bradford-Hill criteria. One of the criteria is whether events are consistently observed in a certain temporal order and, in this work, we study this time concordance using the concept of "first activation" as data-driven means to generate hypotheses on potentially causal mechanisms. As a case study, we analysed liver data from repeat-dose studies in rats from the TG-GATEs database which comprises measurements across eight timepoints, ranging from 3 hours to 4 weeks post-treatment. We identified time-concordant gene expression-derived events preceding adverse histopathology, which serves as surrogate readout for Drug-Induced Liver Injury (DILI). We find known mechanisms in DILI to be time-concordant, and show further that significance, frequency and log fold change (logFC) of differential expression are metrics which can additionally prioritize events although not necessary to be mechanistically relevant. Moreover, we used the temporal order of transcription factor (TF) expression and regulon activity to identify transcriptionally regulated TFs and subsequently combined this with prior knowledge on functional interactions to derive detailed gene-regulatory mechanisms, such as reduced Hnf4a activity leading to decreased expression and activity of Cebpa. At the same time, also potentially novel events are identified such as Sox13 which is highly significantly time-concordant and shows sustained activation over time. Overall, we demonstrate how time-resolved transcriptomics can derive and support mechanistic hypotheses by quantifying time concordance and how this can be combined with prior causal knowledge, with the aim of both understanding mechanisms of toxicity, as well as potential applications to the AOP framework. We make our results available in the form of a Shiny app (https://anikaliu.shinyapps.io/dili_cascades), which allows users to query events of interest in more detail.

expression data was derived from the Life Science Database Archive (https://dbarchive.biosciencedbc.jp/en/open-tggates/download.html). The files and code for the Shiny app are deposited in GitHub (https://github.com/anikaliu/DILICascades_App) and Zenodo (doi:10.5281/zenodo.5767783).

**Funding:** AL received funding from and JM was a full-time employee of GlaxoSmithKline (https://www.gsk.com) throughout the study. NH is funded by LifeArc (https://www.lifearc.org). The funders had no role in study design, data collection and analysis, decision to publish, or preparation of the manuscript.

**Competing interests:** I have read the journal's policy and the authors of this manuscript have the following competing interests: AL received funding from GlaxoSmithKline and is a consultant at PharmEnable Ltd. JM was an employee of GlaxoSmithKline throughout the study and is now an employee at UCB. NH is a cofounder of KURE.ai and CardiaTec Biosciences. AB is a shareholder of Healx Ltd. and PharmEnable Ltd., and CSO at Terra Lumina.

## Author summary

Understanding mechanisms from systems-scale biological data is of great relevance in toxicology as well as drug discovery; however how to generate causal hypotheses instead of correlations is by no means clear. In this work, we study the conserved temporal order of events and present an automatable framework to quantify and characterize time concordance across a large set of time-series. We apply this concept to events derived from time-resolved gene expression and histopathology from the TG-GATEs *in vivo* liver data as a case study. We were able to recover known events involved in the pathogenesis of Drug-Induced Liver Injury (DILI), and identify potentially novel pathway and transcription factors (TFs) which precede adverse histopathology. As complementary sources of evidence for causality, we additionally show how time concordance and prior knowledge on plausible interactions between TFs can be combined to derive causal hypotheses on the TFs' mode of regulation and interaction partners. Overall, the results derived in our case study can serve as valuable hypothesis-free starting points for the development of Adverse Outcome Pathways for DILI, and demonstrate that our approach provides a novel angle to prioritize mechanistically relevant events.

## Introduction

Adverse drug reactions are a major reason for compound failure in the clinical trials [1,2] and a significant cause for post-marketing withdrawals. To counter exposing patients to these risks, it is desired to identify adverse events earlier in the individual patient but also in the drug development process. Mechanistic understanding of adverse event pathogenesis is crucial in this regard, i.e. to derive early safety biomarkers or *in vitro* assays. However, current understanding of toxicity is largely incomplete, in particular for complex phenotypes such as organ injury which can usually be caused by a wide range of compounds perturbing the biological system at different points mediated through multiple biological scales and entities [3,4].

Multiple interrelated concepts have been introduced to formalize mechanistic knowledge in the context of toxicity including Adverse Outcome Pathways, AOPs [4,5]. These begin with a molecular initiating event (MIE) which describes the first interaction of the compound with the system, e.g. a target protein, which is then linked to the adverse event (AE) through a causal cascade of key events (KEs) on different biological levels, like activation of cellular pathways or changes in the tissue or organ. Thereby, the Organization for Economic Co-operation and Development [6] published three criteria to evaluate causality between events within AOPs based on the original Bradford Hill considerations established in the context of epidemiological studies [7] and previous work on the related Mode Of Action (MOA) concept [8]: i) Biological plausibility, ii) essentiality of key events and iii) empirical support for key event relationships. Empirical support for a causal relation between events $E_{cause}$, which could be a MIE or KE, and $E_{consequence}$, which could be an AE, is further separated into time concordance ($E_{cause}$ happens before $E_{consequence}$), dose concordance ($E_{cause}$ happens at lower dose than $E_{consequence}$) and incidence concordance ($E_{cause}$ affects a larger population and is hence more frequent than $E_{consequence}$).

Computational approaches can thereby support these predominantly expert- and knowledge-driven mechanistic efforts by prioritizing mechanistically relevant events or by providing additional insight on the relation between an event and a given phenotype. For instance, computationally predicted AOPs (cpAOPs) prioritize plausible events and event relationships as

starting points for expert-driven AOP development by integrating functional and statistical associations between biological entities on different levels [9–11]. In contrast, probabilistic quantitative AOPs (qAOPs) provide additional insight on the predictivity of key event relationships (KERs) by aiming to predict the adverse event from *in vitro* assays implementing the expert-curated AOP as scaffold [12,13]. To support increasing implementation of *in vitro* methods, however, it needs to be better understood which readouts describing molecular or cellular effects are indicative for systems-level adverse effects in the first place.

Biological readouts such as transcriptomics are particularly suited to study such intermediate key events as they provide broad insights into cellular changes, e.g. in contrast to target profiling, which can then lead to the identification of predictive signatures and mechanistically relevant insights. This is for example true in the context of DILI [14–18], which is a major cause for attrition in drug development and accounts for around half of the cases of acute liver failure in the US and European countries [19,20]. In this regard, in particular time-series data is interesting as it is able to trace the dynamic effects throughout pathogenesis. Previous studies focussed on the time (and dose) dependence of gene expression-derived events in the context of adverse findings [21,22], so the changes of individual events across changes in time (and dose), and also aimed to predict later adverse findings from fixed early timepoints [14,15]. From a mechanistic perspective, however, neither activation at a certain timepoint nor a certain progression over time is mandatory, but only time concordance, so activation of the key event before the downstream adverse effects.

In this study, we hence quantify time concordance across gene expression- derived cellular events and adverse events based on histopathology across a wide range of compounds. To do so, we introduce the concept of "first activation" for mechanistic analysis, which focusses only on the earliest timepoint an event can be reliably detected and then orders events within a time-series by their timepoint of first activation (Fig 1A). In contrast to previous time concordance analyses in AOPs which addressed a defined set of KER and known KE [23–25], this analysis derives statistical evidence for temporal concordance across time-series and can do so for any combination of events based on gene expression or histopathology. Although the confidence of these temporal orders per time series is limited by the noisiness of gene expression data and the low time resolution, statistical significance can be evaluated across time series and we only consider relations time-concordant if the preceding event is significantly more frequently found before the later event than in time series without it (Fig 1B). Furthermore, this also allows us to separate out events which depict general perturbation response but are unspecific, as well as rare events, which are predictive but only observed for a small subset of compounds.

We demonstrate the utility of this concept in this work using liver gene expression and histopathology data from repeat-dose studies in rats provided by the TG-GATEs database (S1 Fig). This allows us to take advantage of previous data curation and work on the dataset itself, in particular by Sutherland et al. [15] who provide an adverse classification of each compound-dose combination and toxscores summarising histopathological findings in each condition. Furthermore, Drug-Induced Liver Injury (DILI) is well understood in comparison to other organ-level toxicities and we hence know which processes are expected to precede injury, including cell death, inflammation and other adaptive stress responses [19]. The concept, however, is generally applicable beyond the toxicity area and transcriptomics data and can be used to derive mechanistic event cascades from time-series data of any kind as long as the first activation of events within the time-series can be defined.

We first describe the time concordance for known processes, similar to mechanistic qAOPs, and then prioritize predictive, time-concordant KE providing a strong data-driven, automatable starting point for AOP development, aligning with the objective of cpAOPs

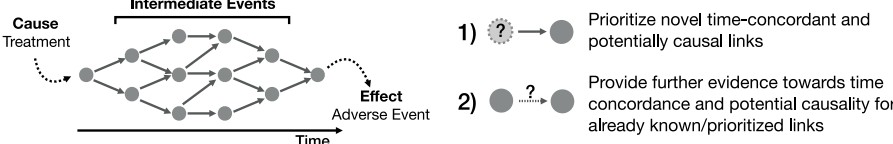

**Fig 1. Quantifying time concordance based on first activation.** (A) The event activation of the events A-D and the later event is shown over time, as well as their timepoint of first activation, at which the event first passes the defined activation criteria. If an event takes place before a defined later event, which in our study is adverse histopathology, it is time-concordant. Time concordance indicates that there is potentially a causal relation between both events, and is distinct from time-dependence which is defined based on the correlation to the later event or time. (B) Based on the frequency of an event before or at the same time as the later event and its frequency in background time-series without the later event, a confusion matrix and different time concordance metrics can be derived. (C) Time concordance can both prioritize novel links and provide further evidence on potential mechanistic links between events. Events are indicated as nodes and mechanistic links between them as edges.

(more detailed comparison in S1 Table). We then combine data-driven time concordance and prior knowledge on event relations between transcription factors (TFs) and gene expression to generate hypotheses for causal gene-regulatory mechanisms in DILI pathogenesis and to generally show how time concordance can stratify and support other streams of causal evidence. Overall, we show that time-resolved gene expression and histopathology data can be used to quantify time concordance across a large set of compounds and events, which allows us to characterize known mechanistic links and to prioritize new ones (Fig 1C).

## Results and Discussion

In order to derive the time concordance between cellular events and later adverse histopathology, we use the workflow outlined in Fig 2 with each step being also introduced in the subsequent sections and details on their respective implementation being described in Methods. We first derived TF and pathway activity across expression profiles from the same experiment and subsequently defined the first up- or downregulation TFs or pathways as events. Furthermore, we obtained binary histopathology labels describing the occurrence of each histopathological finding at different levels of severity and frequency from the toxscores provided by Sutherland et al. [15]. Subsequently, we derive the earliest timepoint of each event, e.g. pathways or adverse histopathology, within each time-series. As last step of the time concordance analysis, we then evaluate which gene expression-derived events are significantly enriched before or at the time where adverse histopathology is found, as well as additional time concordance metrics outlined in Table 1.

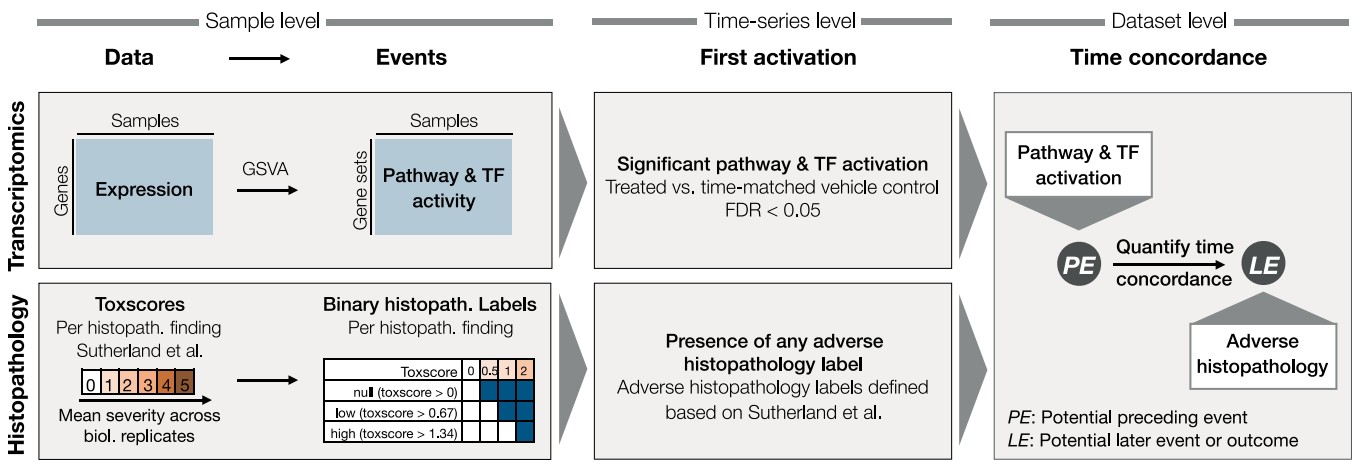

**Fig 2. Workflow to quantify time concordance between preceding gene expression-derived events and later adverse histopathology.** First, events are derived from the gene expression and histopathology data. Pathway and TF activity is inferred based on the expression of the respective gene sets using GSVA [26] and binary histopathology labels are derived from the continuous toxscores. Secondly, we derive the first activation of expression-based events as well as of adverse histopathology. Lastly, we quantify the time concordance between potential preceding events (PE) which are derived from gene expression and adverse histopathology as potential later event (LE).

## Adverse histopathological findings and their temporal relation

To define the earliest timepoint of adverse histopathology within each time-series, we used the annotations of time series as adverse or non-adverse by Sutherland et al. [15], as well as the toxscores, which summarise the severity and frequency for each histological finding and each compound-dose-time combination as mean severity score and range from 0 (normal) to 4 (severe). These toxscores were used to define three levels for each histological finding: "null" (toxscore > 0), "low" (toxscore > 0.67) and "high" (toxscore >1.34). For example in case of a toxscore of 1, both "null" and "low" are considered to be present. We then evaluated which histology groups were frequently found in the adverse compound-dose combinations (observed in >10% of adverse time series corresponding to at least 5 out of 40 cases) with at least 50% of findings being in adverse time series (Fig 3A). All of the included histology groups are significantly enriched in adverse conditions, however, these criteria were implemented to identify findings with a certain specificity and frequency instead of allowing a trade-off between both. The histology groups which passed the filtering are regarded as adverse histopathological findings and include hepatocellular single cell necrosis and biliary hyperplasia at all toxscore thresholds. In contrast, only some of the three toxscore thresholds were selected with the above criteria for all other findings, e.g. the two higher toxscore cut-offs for hepatocellular

**Table 1. Metrics quantifying the time concordance between a potential preceding event *PE* and potential later event *LE*, and their relation to the original Bradford Hill (BH) considerations.**

| BH consideration | Metric | Formula | Description |
|---|---|---|---|
| Consistency | True positive rate (TPR) | p(*PE →LE*|*LE*) | Fraction of time series with event *PE* with specified temporal relation among time series with event *LE* |
| Specificity | Positive predictive value (PPV) | p(*PE→ LE*|*PE*) | Fraction of time series with event *LE* with specified temporal relation among time series with event *PE* |
| Temporality | p-value | One-sided Fisher's Exact test | Likelihood of observing event *PE* and *LE* with specified temporal relation with equal or higher frequency by chance assuming a hypergeometric distribution. |
| Strength | Effect size in time series with *LE* | Median (logFC) | Median logFC of *PE* observed in time-series with *LE* (in comparison to vehicle control) |

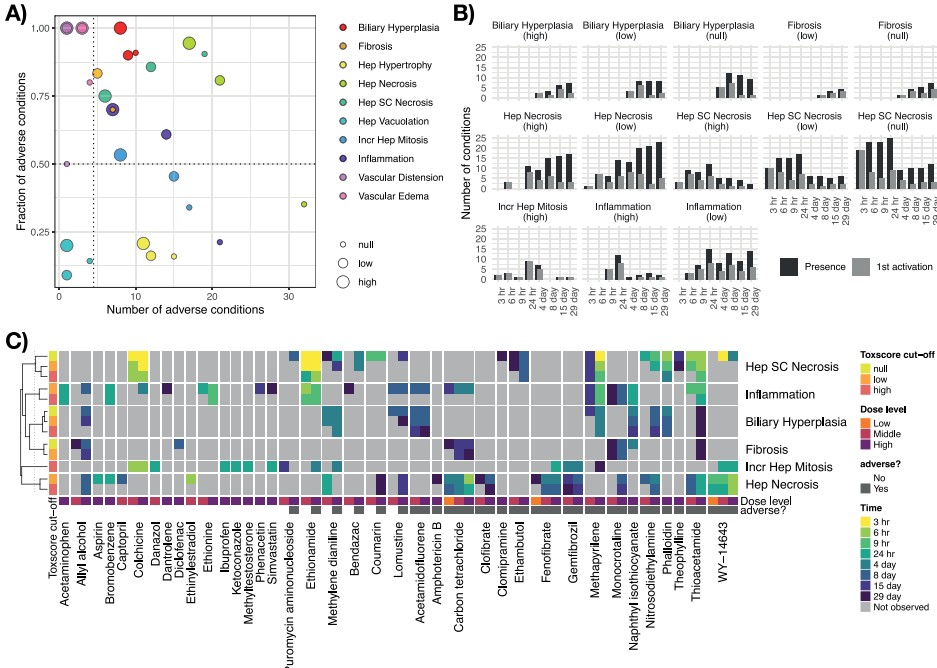

**Fig 3. Distribution and relation of histopathological findings across time series.** A) Histopathology labels are defined for each histopathological finding at 3 different toxscore cut-offs, namely "null" (toxscore>0), "low" (toxscore>0.67) and "high" (toxscore>1.34). For each label, the number of occurrences in the 40 adverse time series and the fraction of adverse time series among all occurrences of the given histopathology label are shown. Histopathological findings, out of which at least 50% and at least 5 of the occurrences were found in adverse conditions timeseries were considered adverse B) Number of conditions with histopathological findings at different timepoints, as well as the frequency of the respective first activations C) Time of first activation across timeseries labelled as adverse or non-adverse. Each time series is annotated with the dose level in repeat-dose studies, as well as with whether or not the time series was considered adverse by Sutherland et al. [15].

necrosis and inflammation and only the "high" cut-off for increased hepatocellular mitosis. In all cases, the lower toxscore level was also frequently observed in non-adverse conditions and hence considered too unspecific. In contrast, only the two milder levels of fibrosis were included in the selection, as severe fibrosis was observed rarely. While we focus on the described definition of adverse histopathological findings in this study, the difficulty in summarising a complex phenotype such as DILI into a binary classification, adverse or not adverse, is well established [27,28] and is also demonstrated by the discrepancies between DILI classifications from DILIst [29], DILIrank [30] and those derived by Sutherland et al.[15] based on the TG-GATEs data (S2 Table). We are aware that also broader or more targeted phenotypes might be of interest, and we hence provide a Shiny app where results for alternative definitions of adverse and non-adverse histopathology groups can be explored.

For the adverse histopathology labels, the distribution of toxscores and first activation over time (Fig 3B) shows that some findings are predominantly found late, like fibrosis, while others are predominantly found early, e.g. hepatocellular single cell necrosis. Next, out of all 360 time-series with at least 6 measured timepoints, the 61 time-series in which any of the adverse histopathology labels is found were identified, which covered 38 compounds (S2 Table). In those, the earliest evidence of an adverse phenotype is used to approximate the timepoint of the primary adverse phenotype. Across all timeseries with adverse histopathology, we find that hepatocellular single cell necrosis is most frequently the primary adverse phenotype, while biliary hyperplasia at any severity is in most cases a secondary effect (Figs 3C and S3).

## Known pathways in DILI preceding adverse histopathology

To identify cellular mechanisms in the early pathogenesis of DILI, we next studied time-concordant cellular changes preceding later adverse histopathology (see Methods). This identified 911 pathway-level events (37.3%), and 108 TF-level events (33.6%) with significant enrichment (p-value<0.05) before or at adverse histopathology. We next evaluated time concordance for a set of ten known events in DILI (Fig 4 and S3 Table). Recycling of bile acids and salts was the most significantly enriched geneset overall and hence also among the ones linked to known events. Also down-regulation of the other bile acid gene sets was significantly enriched (p-value < 0.05) pointing to an overall down-regulation of bile acid metabolism. While cell death was also only found to be up-regulated, dysregulation in both directions was found to precede injury for all other key events (Fig 4). However, only for peroxisomal processes, namely peroxisomal protein import and beta-oxidation of very long fatty acids, both directions were significantly enriched indicating that dysregulation in either direction might be linked to injury. Overall, significantly enriched gene sets are found for all ten represented known events in DILI (p-value < 0.05) indicating that our analysis is able to recover known cellular events.

To gain insights on a more fine-grained level, we next analysed the enrichment of significantly and strongly (absolute log fold change > 1) dysregulated individual genes from the above gene set (S2 File). Among the ten most significantly enriched gene-level events, three are involved in known processes, namely the up-regulation of Acyl-CoA thioesterase 2 (Acot2), Acyl-CoA thioesterase 3 (*Acot3*) and Carnitine O-Acetyltransferase (*Crat*) which are involved

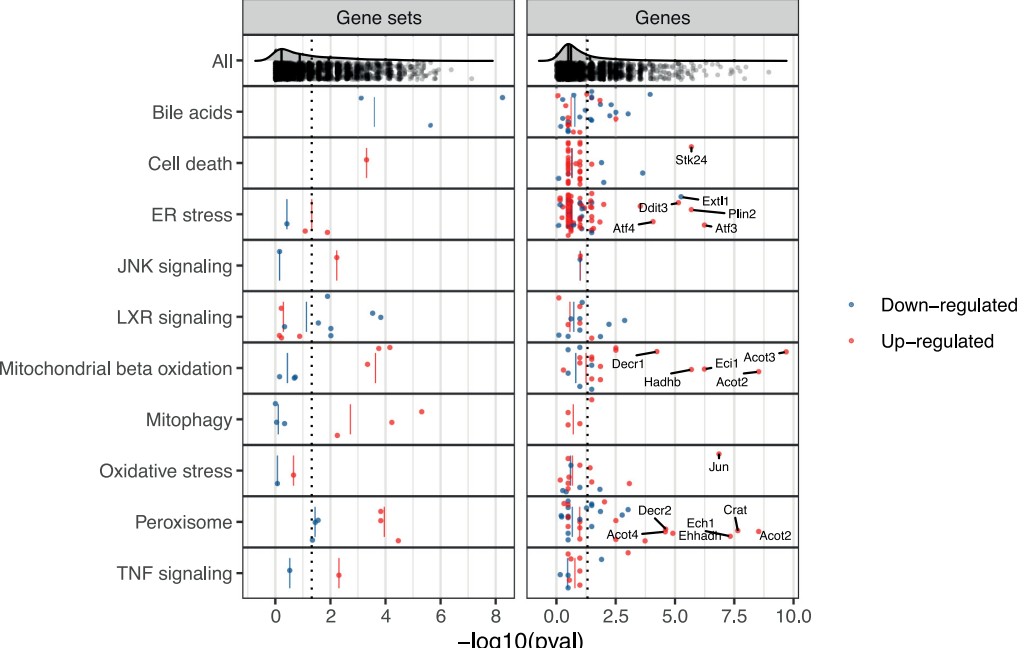

**Fig 4. Enrichment of known events in DILI before adverse histopathology based on gene sets as well as individual gene members.** The enrichment of first activation before or at adverse histopathology is shown for gene sets mapping to known key events in DILI, for which first activation was defined as first timepoint of differential GSVA-derived gene set activity. Furthermore, also enrichment of individual genes within these genesets is shown and was derived based on the first timepoints of differential expression. Aligning with the expected direction, a significant down-regulation of Liver X Receptor (LXR) signalling and bile acid-related pathways is observed, while all other gene sets were found to be more significantly up-regulated. Only for peroxisomal pathways, both directions were significantly enriched indicating that dysregulation in direction might be linked to adverse histopathology.

in fatty acid beta oxidation [31,32]. Multiple genes among the ten most significantly enriched gene-level events are also involved in mitochondrial and peroxisomal processes except for *Gadd45a*, Growth Arrest And DNA Damage-Inducible Protein which has a known role in hepatic fibrosis [33], Neutral Cholesterol Ester Hydrolase 1 (*Nceh1*) which is involved in cholesterol metabolism in macrophages [34], Ras-Related Protein Rab-30 (*Rab30*) which is elevated in early liver regeneration [35], as well as the Serine/Threonine Protein Kinase NIM1 (*Nim1k*).

For JNK signalling, we did not find any significantly enriched genes indicating that while the overall process is changing none of the individual genes shows strong and frequent expression changes. In contrast, the opposite was found for oxidative stress with the Jun Proto-Oncogene (*Jun*) being one of the most significantly enriched gene-level events but lacking significant changes on the gene-set level. This shows that both gene- and gene-set level analysis can provide complementary insights into cellular changes preceding DILI, and that in some cases effects can be attributed in individual genes which might give more detailed information about the cellular changes.

While significant enrichment before or at adverse histopathology can be regarded as a necessary criterion for time concordance, the temporal event relationship can be further characterised based on the observed behaviour across experimental conditions which may be useful to further prioritize mechanistically relevant pathways in a hypothesis-free manner. Following the Bradford-Hill considerations, we hypothesize that this might be the case for observed effect size, frequency and specificity of event occurrence before adverse histopathology. Firstly, we investigated how strongly pathways were dysregulated comparing the maximal absolute log fold changes (|logFCs|) before or at adverse histopathology in each adverse time-series for significantly time-concordant events (Fig 5). High median maximal |logFCs| were overall found for mitochondrial and peroxisomal pathways and the highest median maximal |logFC| among all significant events was found for mitochondrial fatty acid oxidation of unsaturated fatty acids. At the same time, however, the high variance for pathways with high median maximal |logFC| as well as the only moderately high |logFC|s observed for other known pathways in DILI, such as programmed cell death. This indicates that a high magnitude of |logFC| is not necessary to contribute to an adverse event, but at the same time can be a useful property to further prioritize important pathways.

We next analysed to what extent dysregulation in a pathway is predictive for later adverse histopathology. To this end, we calculated across how many adverse time-series each pathway is observed, summarised by the true positive rate (TPR), and the positive predictive value (PPV) indicating whether presence of the key event is a confident indicator for the later adverse event (Fig 6). We focus on significantly enriched events only (p-value < 0.05) and find a trade-off with respect to the highest TPR and PPV (Fig 6; for distribution of all events see S4 Fig). This generally shows that either highly frequent events with lower specificity can be identified, e.g. increased mitophagy (TPR: 0.41, PPV: 0.72), or more specific events at the expense of lower relative frequency, e.g. bile acid recycling (TPR: 0.30, PPV: 1).

Surprisingly, lower relative frequencies are particularly observed for stress response and signaling pathways with only Liver X Receptor (LXR)-dependent gene expression linked to lipogenesis reaching a TPR over 20%. One explanation for the lower observed frequencies is that these pathways are predominantly and initially mediated through post-transcriptional alterations instead of gene expression changes [36,37], making the expression of pathway members a weak proxy for pathway activation in early pathogenesis and explaining the overall low frequencies. In fact, one reason LXR-dependent changes might have achieved higher frequencies might be that they explicitly include the downstream regulated genes unlike the other signalling and stress response pathways [38].

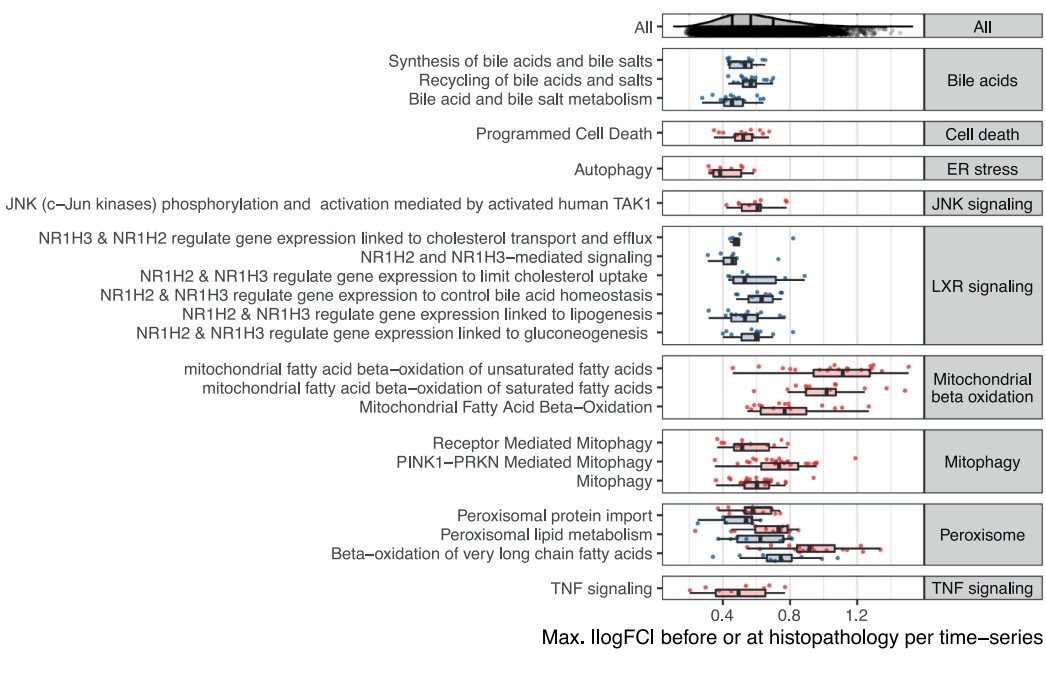

**Fig 5. Observed max. |logFC| before adverse histopathology.** For known processes in DILI which correspond to significantly enriched events before adverse histopathology, the max |logFC| before adverse histopathology is shown. In comparison to other known pathways and the overall background distribution, a high logFC is found for mitochondrial beta oxidation followed by peroxisomal beta oxidation and mitophagy.

Due to the previously discussed complementarity of gene- and gene set-level analysis, we also show TPR and PPV for individual genes with a focus on those which are involved in gene sets mapping to known key events. The most significant genes, already highlighted in Fig 4,

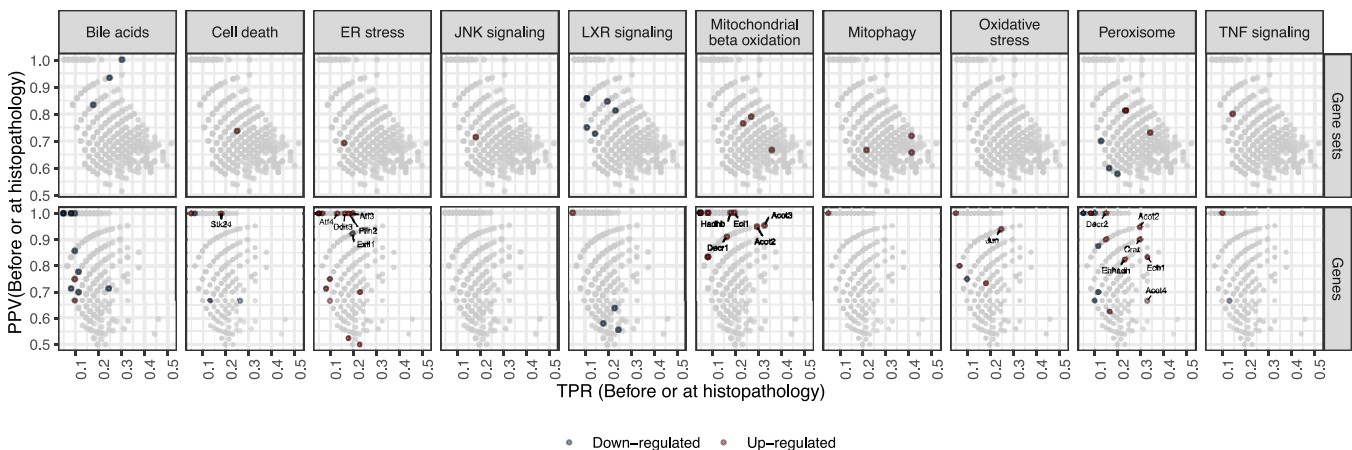

**Fig 6. True positive rate (TPR) and positive predictive value (PPV) before or at histopathology of genes and gene sets in known key events in DILI.** Events related to the given known key event are shown in red or blue indicating an up- or downregulation, respectively. Genes with a p-value < 0.0001 involved in known key events in DILI are additionally labelled. The background distribution of all significantly enriched genes or gene sets is shown in grey (p-value < 0.05).

reveal a high frequency for the up-regulation of the Acyl-CoA thioesterases *Acot2*, *Acot3* and *Acot4*, as well as the for the Enoyl-CoA Hydratase 1 *Ech1* which aligns with the relatively high frequency of pathway-level events linked to mitochondrial and peroxosomal processes. Furthermore, the most frequent gene-level events with a PPV = 1 are the up-regulation of Activating Transcription Factor 3 (*Atf3*) which was found to promote hepatic fibrosis [39] and Enoyl-CoA Delta Isomerase 1 (*Eci1*).

## Known TFs in DILI preceding adverse histopathology

To gain insight into signalling and expression regulation preceding adverse histopathology, we next analysed transcription factors (TFs), which are involved in early perturbation response preceding downstream gene expression changes and also are likely to show strong signal in transcriptomics data given their direct link to gene expression. As known TFs in DILI, we thereby included TFs mediating the stress response and signalling pathways already introduced above, as well as nuclear receptors which take in important roles in liver physiology and malfunctions and can be, both, MIEs or KEs (mapping shown in Fig 7A). Consistent with the pathway-level results, an enriched up-regulation was found for Nuclear factor erythroid 2-related factor 2 (Nfe2l2) which is a key mediator of oxidative stress [18,40] as well as for the Nf-κB subunits Rela and Nfkb1 indicating inflammation [41], while the Oxysterols Receptors LXRα (Nr1h3) and LXRβ (Nr1h2) which control lipid metabolism showed enriched down-regulation [42].

For ER stress, we included three TFs mapping to the three branches of unfolded protein response [43]: Activating transcription factor 4 (Atf4), Activating transcription factor 6 (Atf6) and X-box binding protein 1 (Xbp1). Atf4 up-regulation was found to be most significantly enriched, most frequent and also showing the largest logFC (Fig 7). This highlights its overall importance in mediating ER stress and is consistent with the known role for ATF4 in DILI [44]. While Atf4 is a member of the pro-apoptotic unfolded protein response branch, the Atf6 and Xpb1-mediated branches tend to be cytoprotective [45]. In agreement with this, Atf6 was

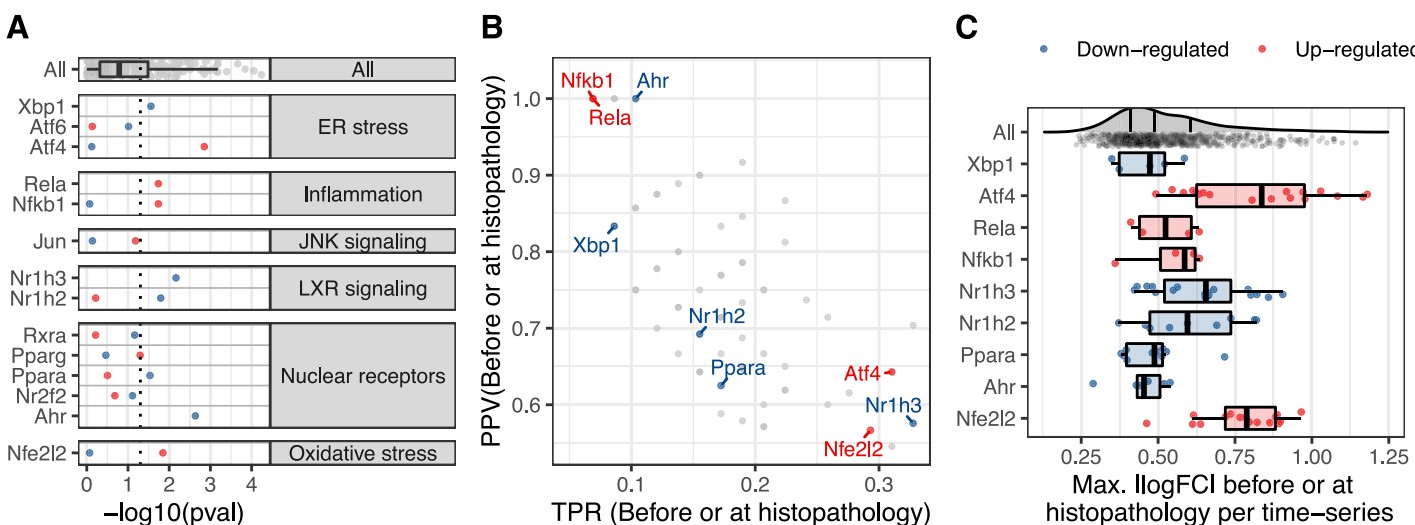

**Fig 7. Temporal concordance of nuclear receptors and adaptive response transcription factors (TFs) in DILI.** For known TFs in DILI the following time concordance metrics are shown: A) The enrichment significance before or at first adverse histopathology, B) Positive Predictive Value (PPV) and True Positive Rate (TPR), C) Max. mean |logFC| before or at first adverse histopathology. As background distribution in grey, the statistics for all inferred TFs is shown.

not significantly enriched, however, Xbp1 showed rare but significantly enriched down-regulation.

Transcription Factor AP-1 (Jun), which is one of downstream target TFs of c-Jun N-terminal kinase (JNK) signaling, was not significantly enriched in either direction due its rare activation among adverse time series although JNK signaling up-regulation itself was significantly enriched with *Jun* up-regulation being one of the most significantly enriched gene-level events. However, JNK signaling is particularly known in acetaminophen-induced liver injury and in this context leads to hepatocyte death through interactions with Sab on the mitochondrial outer membrane and not through transcriptional regulation mediated by AP-1 [46,47]. As increased Jun activity is hence known to be a consequence of JNK signaling but not a cause of injury, it would be plausible to see enriched pathway activity but not in TF activity before adverse histopathology. Overall, we were hence able to show significant enrichment of some of the known TFs in DILI before adverse histopathology and can also biologically reason the absence of significance for others.

While none of the included TF-level events ranked as most significant or most strongly changing before adverse histopathology as in the analysis of pathway-level events, the down-regulation of Nr1h3, which is involved in lipid metabolism, was identified as most frequent event (Fig 7B) indicating that the linked physiological changes are commonly but not specifically found before injury. Similarly, the up-regulation of stress response, indicated by Nfe2l2 and Atf4, was found to be frequent aligning with their role in adaptive stress response [48]. Overall, frequency might hence be a useful metric to identify pre-adverse cellular events which precede injury but are not highly specific.

## Data-driven prioritization of cellular events taking place before adverse histopathology

As many events were found to be significantly enriched before adverse histopathology, we next aimed at identifying and characterizing events most supported by time concordance, and hence at moving closer to the eventual aim of constructing AOPs from data. In our analysis, some known events in DILI ranked highest by enrichment p-value while others rank highest by max. |logFC| before adverse histopathology. In contrast, known TFs in DILI were found as most frequent ones in the dataset. We hence next looked into the top 10 TF- and pathway-level events identified using max. |logFC|, the enrichment p-value, and the TPR before or at adverse histopathology. These are shown in Fig 8 and S4 Table, while all time concordance metrics are summarised in S2 File. The most significantly enriched pathway-level event is decreased bile acid and salt recycling and also the down-regulation of multiple metabolic pathways, in particular targeting glycosaminoglycans, is found among the most significant pathway-level events pointing towards reduced liver function. Moreover, the most significantly enriched TF-level event was the down-regulation of Transcription factor activating enhancer binding protein 4 (Tfap4) which shows emerging roles in cell fate decisions [49], and is followed by Homeobox B13 (Hoxb13) for which expression has previously been found to correlate with hepatic inflammation in hepatic fibrosis [50].

Among the up-regulated events, the most significant enrichment is found for cell cycle checkpoints and DNA repair among the pathway-level events as well as E2F Transcription Factor 2 (E2f2), which controls cellular proliferation and liver regeneration [51], and was found among the most significant TF-level events. E2f2 up-regulation was also identified as 2nd most frequent TF event after the down-regulation of Nr1h3 and among the top 10 most strongly changing TF events further highlighting its strong time concordance. The most frequent genesets point to translation regulation via Eukaryotic translation initiation factor 2A

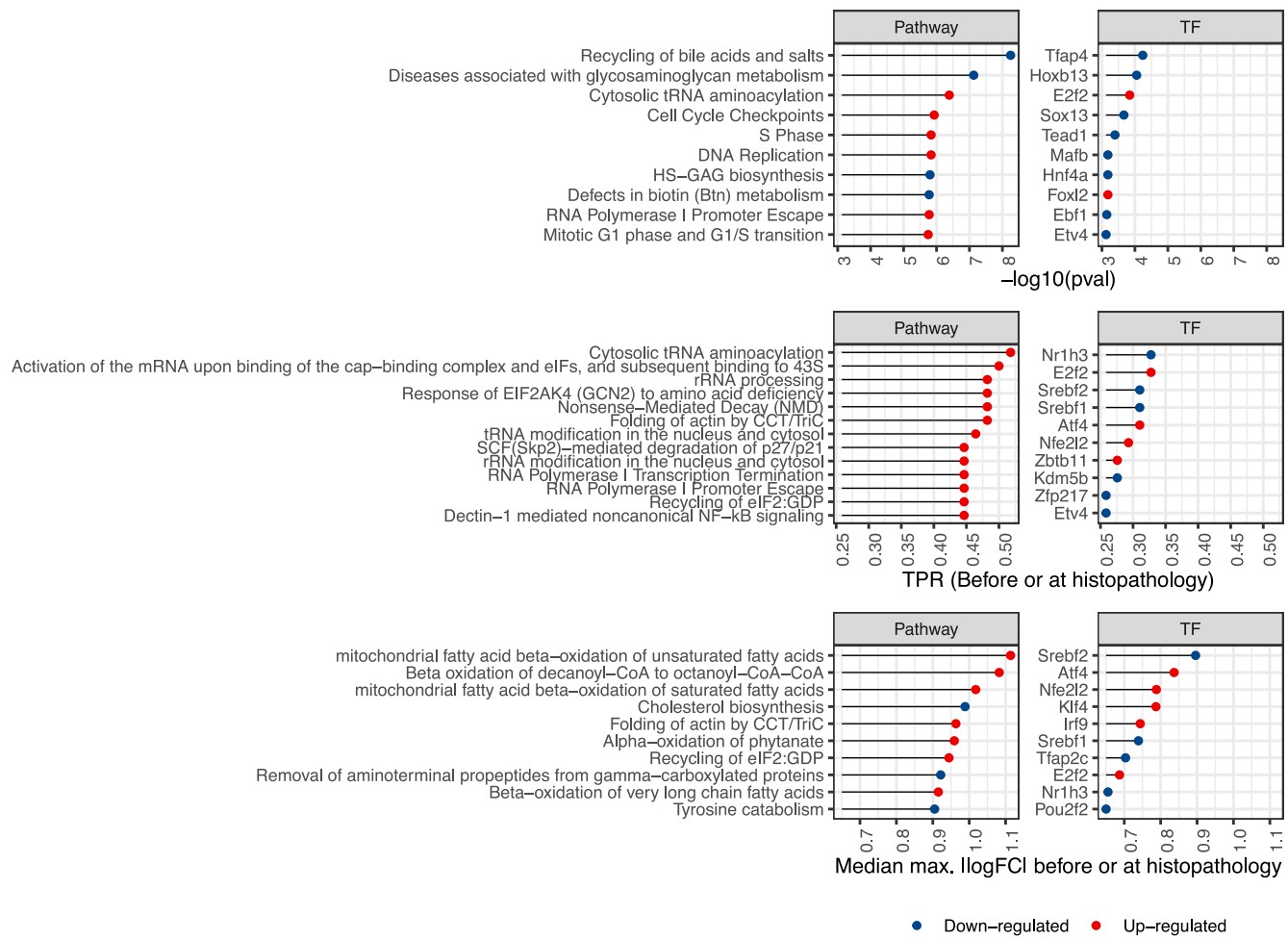

**Fig 8. Highest ranking events by time concordance metrics.** The ten transcription factor (TF)- and pathway-level events ranking highest by enrichment p-value, median max. |logFC| and true positive rate (TPR) before or at histopathology are shown.

(EIF2a) including the upstream response mediated by eIF-2-alpha kinase GCN2 and the downstream role in protein translation mediated through interactions with tRNA. EIF2a is part of the same branch of Unfolded Protein Response (UPR) as Atf4 and causes its preferential translation which, among others, mediates autophagy and proapoptotic response [43,52] and is a known predictor of DILI [44]. Furthermore, increased folding of actin by Chaperonin containing tailless complex polypeptide 1 (CCT) or tailless complex polypeptide 1 ring complex (TRiC) is found frequently and with large effect size. It has been previously linked to proteostasis and autophagy [53,54], but a role in DILI specifically is not yet known. As most strongly dysregulated events, metabolic pathways are found pointing to increased beta oxidation of fatty acids, as well as decreased cholesterol biosynthesis and tyrosine catabolism. Also the most strongly down-regulated TFs point towards lipid metabolism, i.e. the Sterol Regulatory Element Binding Transcription Factor 1, Srebf1, and the Sterol Regulatory Element Binding Transcription Factor 2, Srebf2, as well as Nr1h3 which controls *Srebf1* expression. Overall, the derived time-concordant events, which take place between the beginning of treatment and onset of adverse histopathology, hence include known and plausible events in liver injury which can be further characterized based on their frequency, significance and logFC.

## Identifying mechanistic hypotheses combining known TF functions and time concordance

While both pathways and TFs constitute interpretable events in this study, further prior knowledge is available on how TFs can function on a molecular level allowing us to derive more detailed hypothesis. Firstly, TF activity can generally be modulated through changes in expression or in post-transcriptional regulation as consequence of cellular signaling or environmental changes. In case of transcriptional regulation, changes in mRNA levels should precede changes in TF activity estimated based on regulon expression and hence time concordance can be used to gain support for transcriptional TF regulation. Being only interested in TF events with a potential mechanistic link to liver injury, we studied how significantly concordant expression and activity for each TF are enriched before adverse histopathology. The strongest evidence for a role in DILI pathogenesis is found for 18 TF events which show both significantly enriched TF expression and regulon activity, providing complementary evidence of TF importance and hinting at transcriptional regulation (Fig 9A). While this is not the case for the 17 TF events which only show significantly enriched TF activity but insignificant enrichment of differential expression, including increased E2f2 activity, this can be explained by post-transcriptional regulation potentially describing earlier response patterns which are a direct consequence of upstream signaling. In contrast, 35 TF events with only significant gene expression, such as increased Jun or Myc, might be already showing changes in expression but not sufficiently large changes in activity yet. As this rather indicates a role in later pathogenesis and expression is only regarded as supporting evidence, these TFs have not been included in the next analysis steps.

To derive stronger mechanistic evidence for induction, we next evaluated how frequently expression changes precede TF activity in the same adverse time-series and compare this against the overall frequency of TF event occurrences preceding adverse histopathology (Fig 9B). Among the events with significant enrichment of TF expression and activity, the most

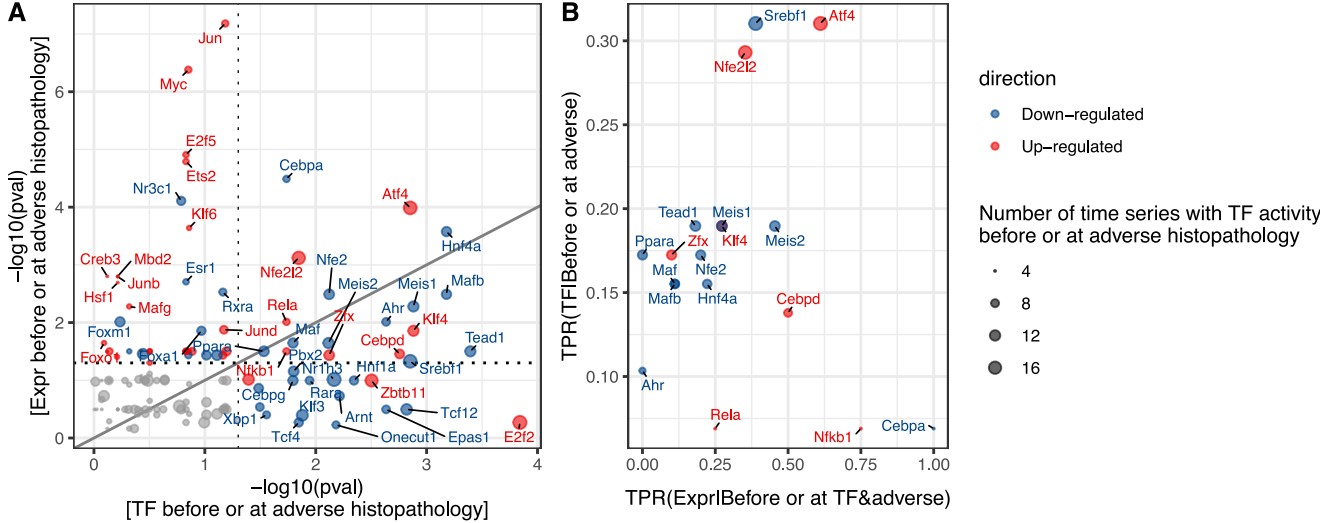

**Fig 9. Transcription Factor (TF) activity and expression before adverse histopathology.** A) Significance of enrichment in adverse conditions for matched TF activity and expression-based events. Events only found on the expression or TF level are not included in the figure due to the inability to perform a statistical test for those. B) For significantly enriched TF activity-based events, the True Positive Rate (TPR) of observing TF activity before or at the time of adverse histopathology is shown, as well as the TPR for observing TF expression changes before TF activity in the time series where it precedes adverse histopathology.

frequent evidence for induction was found for the down-regulation of CCAAT/enhancer-binding protein alpha (Cebpa). In humans, decreased expression of the homologous CEBPA is not only known across liver diseases, exogenously increased CEBPA expression has also been shown to reverse liver injury and is explored as therapeutic target in hepatocellular carcinoma [55]. The event with the 2nd highest relative frequency of expression preceding TF activity as well as the highest frequency of TF activity preceding injury is Atf4, for which expression of the homologous gene in humans is known to be induced as part of the ER stress response contributing to adverse liver phenotypes [56,57]. In contrast, it was found that for the Aryl Hydrogen Receptor (Ahr) and the Peroxisome Proliferator-activated Receptor Alpha (Ppara) changes in expression never preceded those in TF activity which aligns with their roles as nuclear receptors which are generally post-translationally activated via ligand binding [58,59]. As this provides counterevidence for transcriptional induction, these were not included as induced TF in the subsequent analysis.

After investigating the mode of regulation for individual TFs above, we next considered how these TFs are interlinked. To this end, we identified protein-protein interactions and, for induced TFs, TF-target gene interactions between significantly enriched TFs, which showed significant enrichment before adverse histopathology for both expression and regulon activity, as well as evidence of expression preceding TF activity within the same adverse time series. Results of this analysis are shown in Fig 10, and details on the observed absolute and relative frequencies, as well as the source of the interaction are shown in S5 Table. One of the two identified interactions by highest absolute frequency is Nr1h3 down-regulation resulting in reduced Srebf1 activity. Furthermore, Srebf1 is also linked to upstream regulation by Nr1h2 which interacts with Peroxisome Proliferator-Activated Receptor (Ppara) in both directions, and this cross-talk between Ppara and LXR regulating Srebf1 expression has been explicitly studied in the context of fatty acid metabolism regulation [60–62]. The 2nd most frequently observed interaction is the down-regulation of Transcription Factor 12 (Tcf12) inducing reduced activity of TEA Domain Transcription Factor 1 (Tead1). While Tead1 is indeed known to be involved in liver diseases and injury [63,64], the interaction itself has not been reported before in the context of liver injury and the same applies also for the other upstream Tead1 regulators identified. It should also be noted that for these interactions first activation is only found at the same time but not in the time-concordant order providing weaker evidence than, for example, the interaction between Nr1h3 and Srebf1. As additional larger TF cluster, decreased activity of the Hepatocyte Nuclear Factor 1 (Hnf1a), Retinoic Acid Receptor alpha (Rara) and Pancreatic And Duodenal Homeobox 1 (Pdx1) was found to lead to decreased expression and activity of Hepatocyte Nuclear Factor 14 (Hnf4a) which is linked to reduced expression and activity of CCAAT/enhancer-binding protein (Cebpa) through edges in both

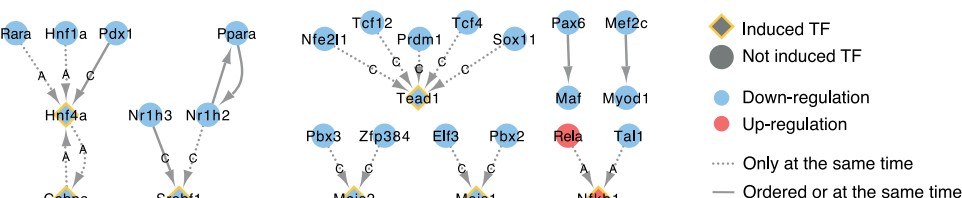

**Fig 10. Causal relationships between TFs supported by time concordance.** For TFs which are significantly enriched before or at adverse histopathology, known causal relations are shown in which the upstream event is found before or at the downstream event in at least 20% of adverse cases. For induced TFs for which expression is found before regulon activity and significantly enriched, not only protein-protein interactions are considered but also upstream TF-target gene interactions annotated with DoRothEA [69] confidence scores (A: High confidence, C: Medium confidence).

directions. This cluster stands out due to the high confidence score of all interactions, except the edge between Pdx1 and Hnf4a, indicating that there is strong support based on prior knowledge for the involved interactions. Furthermore, it was previously found that artificially increased expression of Hnf4a is able to reverse hepatic liver failure in rats, while also restoring expression of a highly interconnected TF network including Hnf1a and Cebpa which supports the identified interactions [65,66]. Two of the yet unknown TFs in DILI are Meis Homeobox 1 (Meis1) and Meis Homeobox 2 (Meis2) which are generally known in a developmental context [67,68]. However, their down-regulation in early pathogenesis is supported by enriched TF activity, differential expression before adverse histopathology as well as upstream regulators which are also enriched before adverse histopathology.

### Time-concordant events reflecting disease progression

While events do not have to be activated continuously to be causally involved in pathogenesis, events with consistent or increasing activation over time are particularly interesting as bio-markers as they can be experimentally measured without the chance of missing the timepoint of activation, and can potentially reflect disease progression beyond early pathogenesis. We therefore studied which TFs and pathways show time-dependent activation by testing for significant Spearman correlation between activation logFC and time in adverse time-series, and whether this overlaps with the previously derived time concordance (Fig 11). Overall, 118 pathways and 19 TFs were supported by both, significant time concordance and dependence, which represents 86.1% or 70.4% of the time-concordant events, and 59.9% or 48.7% of the time-dependent events, respectively.

On the pathway level, multiple genesets pointed to a reduced level plasma lipoprotein particle assembly and remodelling which indicates changes in lipid distribution. This aligns with the known dyslipidaemia in chronic liver diseases, including decreasing serum values of LDL, HDL, total cholesterol, and triglycerides with increasing severity of disease, based on which previous studies suggested that routine monitoring of lipid profiles can improve the outcome for CLD patients [70]. Furthermore, a down-regulation of response to metal ions was found which could be related to metallothioneins which protect against oxidative stress and are able to chelate heavy metals [71]. Both directions of dysregulation were previously observed in liver diseases: While a negative correlation with disease progression was found in hepatocellular carcinoma [72], a positive correlation was found in most other liver diseases including acetaminophen-induced liver injury [73]. This indicates that opposite directionality is more plausible based on current literature knowledge, but cannot be fully clarified. The most time-concordant and -dependent TF event was down-regulation of SRY-Box Transcription Factor 13 (Sox13) which is generally involved in cell fate [74] and embryonal development [75]. As Sox13 does not yet have well understood functions on a more detailed level, experimental validation of a potential role in DILI would be interesting. In contrast, the next most significant time dependence is found for the hepatocyte nuclear factors Hnf1a and Hnf4a, as well as Cebpa, which are known to negatively correlate with liver cirrhosis in rats [76,77]. Overall, this shows that a mechanistic role for time-concordant and -dependent events is strongly supported by the understanding of adverse liver phenotypes.

While, in general, events with highly significant time dependence also showed highly significant time concordance, some exceptions were found in which only one of both was highly significant. For instance, the pathway with the 2nd most significant time-dependence (p-value $< 10^{44}$) is signaling via advanced glycosylation end product receptor (RAGE) which contributes to inflammation and oxidative stress generation and did not pass the significance threshold for time concordance (p-value = 0.058). RAGE expression and activity, which are both induced by

## A) Spearman correlation vs. time concordance significance

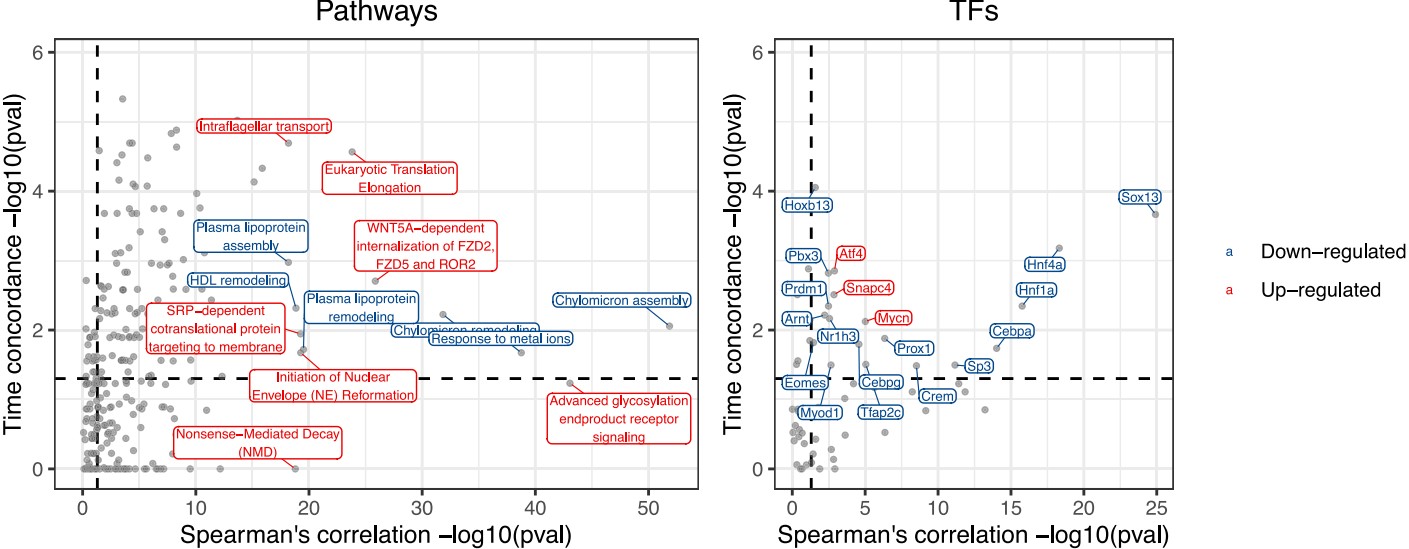

## B) Time-concordant events with most significant Spearman correlation

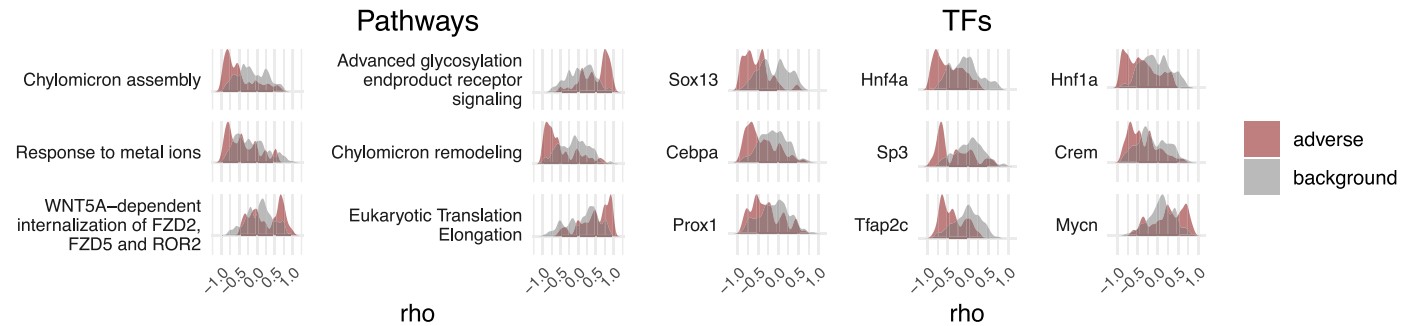

**Fig 11. Combining time dependence and concordance to identify mechanistically supported biomarkers.** A) The relation between time concordance, quantified by the enrichment p-value for event activation before adverse histopathology, and time dependence, quantified by the meta p-value for Spearman correlation between time and event activation across adverse conditions, is shown. B) For events with the most significant time-dependence, the distribution of correlation coefficients is shown providing further insight into the strength of correlation and consistency across adverse conditions.

binding of RAGE ligands, are thereby known to be up-regulated in various hepatic disorders resulting in a positive feedback loop explaining increasing or sustained RAGE activation [78]. This indicates that, while RAGE signaling is correlated with progression, there is no clear evidence for a role in early pathogenesis preceding adverse histopathological changes. In contrast, SUMOylation of TFs, is time-concordant (p-value = 0.002) but not -dependent (p-value = 0.48) indicating a mechanistic role in early pathogenesis which is not sustained over time. This aligns with the finely regulated and pleiotropic roles of SUMOylation in post-transcriptional regulation which have also been found to be involved in the context of liver diseases [79].

## Limitations of this study

In this study, we introduced a time-concordance based approach to derive mechanistic insight from gene expression and histopathology data. We were able to recover known mechanisms in

DILI as well as able to propose novel and detailed mechanistic hypotheses. However, the present analysis is based on a limited number time-series as well as only few timepoints within each time-series. This does not only mean that rare events might be missed as they occur between measured timepoints and that small effects might not be identified as significant, but also that there is potentially a bias based on the tested compounds towards the represented modes of toxicity.

Furthermore, the analysis is limited by how confidently biological processes are inferred from the data. This was for instance demonstrated by the differences between pathway and TF activation for signalling and stress response pathways highlighting the discrepancy between protein activation and gene expression. As only pathways induced through changes in gene expression or their downstream expression footprints [69] can be confidently detected, this means that good estimates of time concordance can predominantly be derived for intermediate or later key events while preceding key events or molecular initiating events which are not mediated by transcriptional regulation cannot be estimated based on the data. That being said, this is a limitation of gene expression data in general and the time concordance approach would also be able to integrate other data types describing events not covered yet.

Moreover, multiple choices were made to align our analysis to the AOP concept prioritizing mechanisms supported by prior knowledge over purely data-driven hypothesis. First, detailed insights might be lost by summarising results to the pathway level. While generally measurements for individual genes can be noisy, this can be summarised in different ways e.g. based on similarity in expression profiles [15]. In this study, however, we used curated gene sets due to their interpretability and to derive modular events as defined in the AOP framework. Additionally, prior knowledge was taken as ground truth, both in the gene set and interaction analysis, meaning that only generally known pathways and interactions could be discovered. Like all methods based on curated gene set and interactions, it was hence informed and biased by the current understanding of biology. However, this prior biological knowledge contributes to the biological plausibility of the derived events and relationships contributing to the weight of evidence of our findings in the context of AOPs.

Lastly, it should be highlighted that time concordance is necessary for causal relations but not sufficient to prove it. For instance, two events may be time-concordant because they are causally linked to a shared preceding cause. To distinguish these effects, the additional Bradford-Hill considerations can be helpful, but only prior knowledge has been considered in parts of this study. In particular essentiality would provide strong evidence for causality, however, requires targeted experiments and hence is unsuitable for hypothesis generation. In contrast, dose and incidence concordance are generally feasible from a data-driven standpoint but were not pursued in this case study due to the low number of doses and replicates.

## Conclusion

In this study, we introduce "first activation" as concept to quantify the strength of temporal concordance between events across time series with the assumption that each activated event may have downstream effects irrespective of whether it is continuously or only transiently activated. With this approach, we study gene expression-based TF and pathway-level events found before adverse histopathology indicating liver injury in repeat-dose studies in rats from TG-GATEs as a case study. We find some known processes in DILI to be highly confident, e.g. bile acid recycling, while others are highly frequent but less specific including adaptive response pathways such as the eIF2α/ATF4 pathway [52].

Beyond quantifying time concordance for known and potentially novel events in DILI, we additionally show how time concordance can be combined with prior biological knowledge to

generate hypothesis on potentially causal gene-regulatory cascades in DILI. Amongst others, this identifies LXRα down-regulation leading to decreased *Srebf1* expression, an interaction known to regulate fatty acid synthesis in the liver [42], but also characterizes yet unknown TFs based on their time concordance, their mode of regulation (either transcriptional or post-transcriptional) and potential upstream regulators and downstream effectors. Two of the identified induced TFs are Meis1 and Meis 2 which are supported by significantly enriched decrease in expression and activity before adverse histopathology, as well as upstream regulators which also show significant enrichment of regulon activity and are found within the same time series. On top of time concordance, we also derive each event's time dependence and show that events mechanistically involved in early pathogenesis do not necessarily reflect disease progression and vice versa. However, for some events, e.g. Sox13, both properties are found and these may be useful biomarkers which reflect injury progression and already change preceding histopathological manifestation.

We believe that the described analysis can provide supporting evidence for mechanistic links between events in line with the evolved Bradford-Hill considerations on time concordance and biological plausibility and can hence e.g. support AOP development. Furthermore, the approach is not limited to a particular adverse event and can instead quantify the interaction between any two events represented in time series in a data-driven and automatable fashion. Consequentially, this type of analysis could also be of interest to study the mechanism of action of particular compound classes or patterns of disease progression. We make the results of our analysis on the TG-GATEs *in vivo* liver data publicly available in a Shiny app through which users can query the most time-concordant events for more specific types of histopathology and study in detail in which time series time concordance was observed or not observed (https://anikaliu.shinyapps.io/dili_cascades).

## Methods

### Open TG-GATES data processing

The TG-GATES gene expression data from studies in 6-week-old male Crl:CD Sprague-Dawley (SD) rats with daily repeat-dosing (S1 Fig) was downloaded from the Life Science Data Archive (DOI: 10.18908/lsdba.nbdc00954-01-000). The raw liver gene expression levels were background corrected, $\log_2$ transformed, and quantile normalized with the rma function of the affy package per treatment across all doses and timepoints [80]. Quality control was then performed using the ArrayQualityMetrics package [81] and detected outliers with high distance to other experiments or unusual signal distribution were removed (List of removed outliers summarised in S1 File). The platform information for the Affymetrix Rat Genome 230 2.0 Array was derived from Gene Expression Omnibus [82] (GEO accession: GPL1355) and was then used to summarise probe IDs to rat gene symbols by median for all probes mapping uniquely to one gene symbol. Only the 360 compound-dose combinations with at least 6 measured timepoints after quality control were included. Out of these, all eight timepoints were measured in most time series, while only six timepoints were measured in two time series, and only seven timepoints in seven time series.

### Definition of adverse histopathology

To characterize the extent of histological findings, we used the toxscores by Sutherland et al. [15] in order to consider both severity and frequency of events in a single numerical output measure. These are based on the lesion severity per animal which was first converted to a numerical scale (normal = 0, minimal = 1, slight = 2, moderate = 3, marked or severe = 4) and then averaged across all biological replicates as an aggregate measure for lesion frequency and

severity. One characteristic of this measure is that the overall distributions varied between different findings, e.g. inflammation was more frequently annotated with low than with high tox-scores while a more balanced distribution of scores was observed for hepatocellular single cell necrosis (S2 Fig).

To study which histological findings were enriched in adverse conditions, we first defined binary histopathology labels describing the presence of histological findings with different extents in each time-series. Based on the toxscore ranges used by Sutherland et al. [15], three toxscore cut-offs are implemented to describe each histopathological finding "Null" (toxscore > 0), "low" (toxscore > 0.67) and "high" (toxscore > 1.34). We then studied which labels were over-represented in adverse time-series. These were defined using the annotation of Sutherland et al. [15], where pathologists classified compound-dose combinations in the TG-GATEs database as adverse or non-adverse after 4 and 29 days of treatment. We used the 29 days classification to define 40 adverse time series and only regarded time-series as non-adverse for compounds which were not classified as adverse at any dose in the negative control, in order to account for the fact that some of the cellular changes of interest might already take place at lower doses, although the resulting phenotype is not considered adverse yet.

We defined findings as adverse histopathology if they are observed in at least 5 out of 40 adverse time-series to remove rare histopathological findings, and additionally require that at least 50% of findings are in adverse conditions to remove findings which are unspecific. All labels which were identified with these criteria are significantly enriched among time-series labelled as adverse by Sutherland et al. [15] in comparison to those that were considered non-adverse using a one-sided Fisher's Exact test (p-value < 0.0001), performed using the *fisher.test* function of the *stats* R package [83]. However, this combination of additional criteria was chosen to exclude findings which are rare or weakly associated.

While not all compounds in the TG-GATEs database are drugs and some mechanisms of toxicity may not translate to humans, out of the 38 compounds represented in adverse time-series, 22 have additionally been classified as hepatotoxic in DILIst [28] and 18 in DILIrank (vMost-DILI--Concern or vLess-DILI-Concern) [27] (S2 Table). This overlap with compound-level DILI annotations by the FDA shows that the compounds in this study partially represent known mechanisms of DILI in humans, while also highlighting the fact that a clear classification is not possible.

## Pathway and TF activity inference

The activity of pathways and TFs across all doses and timepoints of a treatment including vehicle controls was derived based on the expression of its gene set members using GSVA [26], which computes a gene set enrichment by sample matrix from the gene expression by sample matrix. This was performed using a Gaussian kernel requiring at least 5 genes per gene set, and overall provides the basis for the subsequent pathway- and TF-centric steps. As prior knowledge, we used pathway maps from Reactome [38] which were derived through MSigDB [84] and the msigdbr package [85]. TF activity gene sets were derived from DoRothEA [86] and mapped from human to rat gene symbols with biomaRt [87]. These gene sets describe known, functional TF-gene interactions and are assigned a confidence level based on the strength of evidence of these interactions. Thereby, only the 207 TFs with a high to medium confidence level of A-C were included and the few TF-gene interactions with a negative mode of regulation were removed to better infer TF directionality. To evaluate which pathway or TF is dysregulated, we computed the differential activity in comparison to the vehicle control group, which was treated for the same amount of time and as part of the same experiment, using the moderated t-statistic in limma [88]. We identify significantly dysregulated gene sets with a False Discovery Rate (FDR) < 0.05.

## Temporal concordance of events

In this study, the order of events was derived based on each event's timepoint of first activation within each time-series (Fig 1A). For pathways and TFs, we defined first activation as the earliest time of measurement at which significant differential regulation was observed (FDR < 0.05) in each direction, while an additional logFC cut-off has been implemented for individual genes. As first evidence of adverse morphological changes in the liver, we used the first timepoint at which any of the adverse histopathology label derived before were found.

We were then generally interested in potential preceding events *PE* which are first activated before or at the same time as a potential later event or outcome *LE* and used multiple metrics to quantify the degree of time concordance which can be related to the original work by Bradford Hill (Table 1). Thereby, the key later event in this study was adverse histopathology but we used a more general notation *LE*, as some of the following criteria to quantify time concordance are also applied in the TF analysis, where gene expression-derived events are used as later event. First, we used the true positive rate (TPR) which describes how frequently *PE* is observed before *LE* among all time-series with *LE* and hence its consistency across compounds. Secondly, we use the maximal effect size of *PE* observed before *LE*, summarised across time-series by median, to characterise the strength of association. To evaluate the significance of the findings, we additionally defined a set of background time-series unrelated to *LE* (Fig 1B). For adverse histopathology, these unrelated background time-series were the 133 time-series without any observed histological changes. We then computed the enrichment of *PE* before or at *LE* using the *fisher.test* function of the *stats* R package [83], first estimating the odds ratio using the conditional maximum likelihood estimate and subsequently testing the null hypothesis whether the odds ratio derived from a confusion matrix as described in Fig 1 is equal to or smaller than 1. Additionally, we compute the positive predictive value (PPV) of *PE* for *LE*, which describes how likely *LE* is observed at the same or a later time given the observation of *PE.*

Across all metrics, we only consider time-series in the statistics for which any event of the same type as *PE*, e.g. TF or pathway, was observed at the included timepoints, so before or at *LE* or at any timepoint in the background time-series. We do this to account for the fact that in some cases no changes are found which may be a consequence of the fact that there isn't a measured timepoint before *LE* or that at the available timepoints expression changes cannot be detected. We argue that in these cases this should not be treated as evidence of absence of the given event, but rather as absence of evidence.

## Combining time concordance on TF-TF interactions

We used three sources of causal prior knowledge to derive mechanistic hypotheses linking TFs: Protein-protein interaction between TFs derived from Omnipath through OmnipathR [89,90], TF-target gene interactions from DoRothEA [86] and the link between gene expression and protein levels following the central dogma of molecular biology. Using these interactions as backbone, we then derived those additionally supported by time concordance. Thereby, the dysregulation of the nodes was required to match the reported mode of regulation (edge sign) and the source node or upstream event was required to be observed in at least 20% of cases before or at the same time as the target node or downstream event. For induced TFs, significant enrichment of gene expression (|logFC|>0.5) and TF activity before adverse histopathology was required, as well as evidence for changes in expression preceding changes in the same direction in regulon activity within the same time series.

## Time dependence

In each adverse time-series, we tested for Spearman correlation between timepoint and event activation logFC using the correlation R package [91], and include a logFC of 0 at timepoint 0 h assuming that there are no differences in comparison to the control group before treatment. We then identified pathways and TFs which only show significant Spearman correlation in one direction, positive or negative. For those events, we apply the Fisher's combined probability test using the metap R package [92] across all adverse time-series to evaluate whether overall significant correlation between event activation and time is found.

## Supporting information

**S1 Table. Comparison of quantitative Adverse Outcome Pathway (qAOP) models.** Comparison of the first activation concept and other qAOP models with respect to their potential roles in AOP development.
(DOCX)

**S2 Table. Compounds classified as adverse based on histopathology and concordance with previous annotations.** For the annotations by Sutherland et al. [15], who classified each compound at each measured dose as adverse or non-adverse at day 4 and day 29, the adverse doses for each compound are listed. Furthermore, the binary classification as adverse (1) and non-adverse (0) from DILIst [29] are included as well as the vDILIConcern and Severity Class classifications from DILIRank [30] which describe evidence for liver side effects observed in humans derived from post-marketing data.
(DOCX)

**S3 Table. Time concordance metrics for Reactome pathway maps which map to known key events based on literature review.**
(CSV)

**S4 Table. Time concordance metrics for top 10 ranking events by True Positive Rate (TPR), significance and median max. |logFC|.**
(DOCX)

**S5 Table. TF-TF relations supported by known relations and time concordance.** For TF events which are significantly enriched before or at adverse histopathology, known interactions supported by time concordance are shown. With respect to the interaction, the absolute and relative frequency are shown for how often the source TF was observed "before" or "before or at" downstream TF activity. Additionally, the source of the interactions provided in Omnipath are shown for protein-protein interactions and the DoRothEA confidence level for TF-target gene interactions.
(DOCX)

**S1 Fig. Open TG-GATEs study design.** 6-week-old male Crl:CD Sprague-Dawley (SD) rats were treated with a range of compounds using daily repeat-dosing. For each compound, four doses were used including a vehicle control, and samples were taken at 8 timepoints. For each combination of compound, timepoint and dose, histopathology was annotated and gene expression measured for 3 replicates.
(PDF)

**S2 Fig. Distribution of toxscores across histopathological findings.**
(PDF)

**S3 Fig. Frequency of histopathological findings before and after first adverse histopathology.** For adverse and non-adverse histopathological findings, the frequency before or at first adverse histopathology is shown (left). For adverse findings, this indicates how frequently they were one of the first adverse histopathological findings given that they cannot occur before by definition. This identifies single-cell necrosis at any severity ("null"), as the most frequent finding, both in absolute and relative terms.
(PDF)

**S4 Fig. Background distribution of temporal association metrics across pathway and Transcription Factor (TF) events.** The dependency between different metrics is shown. A) Frequency of events by median max. |logFC| before or at histopathology and enrichment p-value. B) Frequency of events by true positive rate (TPR) and positive predictive value (PPV) before or at adverse histopathology. C) Direct relation between TPR, PPV and enrichment p-value. D) Direct relation between TPR, PPV and frequency in background time-series.
(PDF)

**S1 File. Removed outlier samples.**
(CSV)

**S2 File. Time concordance metrics for all TFs, pathways as well as genes using both a minimal |logFC| of 0.5 and 1.**
(XLSX)

## Author Contributions

**Conceptualization:** Anika Liu.

**Data curation:** Anika Liu.

**Formal analysis:** Anika Liu.

**Investigation:** Anika Liu.

**Methodology:** Anika Liu.

**Software:** Anika Liu.

**Supervision:** Namshik Han, Jordi Munoz-Muriedas, Andreas Bender.

**Visualization:** Anika Liu.

**Writing – original draft:** Anika Liu.

**Writing – review & editing:** Anika Liu, Namshik Han, Jordi Munoz-Muriedas, Andreas Bender.

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
