## [Decision Letter · Decision Letter 0]

22 Feb 2022

Dear Ms. Liu,

Thank you very much for submitting your manuscript "Deriving time-concordant event cascades from gene expression data: A case study for Drug-Induced Liver Injury (DILI)" for consideration at PLOS Computational Biology.

As with all papers reviewed by the journal, your manuscript was reviewed by members of the editorial board and by several independent reviewers. In light of the reviews (below this email), we would like to invite the resubmission of a significantly-revised version that takes into account the reviewers' comments.

We cannot make any decision about publication until we have seen the revised manuscript and your response to the reviewers' comments. Your revised manuscript is also likely to be sent to reviewers for further evaluation.

Sincerely,

James Gallo

Associate Editor

PLOS Computational Biology

Mark Alber

Deputy Editor

PLOS Computational Biology

Reviewer's Responses to Questions

**Comments to the Authors:**

Reviewer #1: This manuscript introduces “first activation” as the method to quantify time concordance to provide evidence for causality. The authors perform a case study on a liver dataset from repeat-dose study in rats using the gene expression and histopathology data. This manuscript uses known evidence and prior biological knowledge in DILI to validate the proposed methodology and points out some potential new TF interactions.

I think this manuscript is well-written and organized but could be improved with some analysis aided. I have following detailed comments.

General comments:

1. The key concept of this manuscript is “first activation” and the authors argue that this is generalizable and automatable to other topics. However, the discussion on this concept is quite limited, and should be extended. How the authors defined “first activation” was only mentioned in Methods and should be written in the main text. The threshold seems to be very important as this could lead to distinct results and authors should discuss how robust the current threshold is.

2. The authors mainly used 10 known events in DILI for their case study, but as they mentioned, they identified hundreds of both pathway-level and TF-level events in the datasets. Although using well-known events to validate the methodology is very much needed, analysis on some other less well-studied events as examples should be also discussed.

Minor points:

1. This manuscript focuses on one case study, a liver dataset from repeat-dose studies in rats, a figure showing the concept of this experiment, depicting the info about collecting timepoints, doses etc is helpful and needed.

2. To establish causality, there are multiple rules in Bradford-Hill criteria, while this manuscript focuses only on temporal order, the other rules should be discussed to infer causality.

3. Results that used method described in Method section should be indicated.

4. Line 72, AE was first mentioned and need full descriptions.

5. Fig 1, defining A,B,C,D as candidates preceding events in panel A) causes confusion in panel B), where B is defined as the potential later event.

6. More descriptions needed for S1 Table, not clear what + or ++ means.

7. Reasoning behind choosing 10% and 50% as the threshold needed.

8. Fig2A, unclear about what is Number of adverse conditions vs Fraction of adverse conditions.

9. Fig3 right panel, texts overlap.

10. Fig 4, the figure should be reorganized. The text of ‘Up-regulated’ and ‘Down-regulated’ are unnecessary since there are color codes for it. The order of these ten events is reversed as Fig 3. Why oxidative stress is missing?

11. Fig5 is duplicated 3 times in the manuscript.

12. Fig5, ‘If the event was over- represented before adverse histopathology (p-value < 0.05) the point was additionally circled in black’, very hard to see the black circle in figure, need to change to a different coloring scheme.

13. Fig 5, there are a lot of individual genes are highlighted, but the text did not mention why they were highlighted.

14. The captions and legends of figures in this manuscript need substantial improvement. The audience should be able to get the key ideas from the figure by looking at the figure and caption and legends.

15. Fig 8B, description on the legend should be clearer in the captions. For example, ‘active_tf’ should be described. Also the axis label ‘TPR(Expr|Before or at TF&adverse)’ needs description.

16. The upper or lower case of term (e.g. TF) should be kept consistent between figures and texts. For example, Line 373, ATF4 while in Fig 8B it’s Atf 8. There are many other places having this inconsistency.

17. Line 375, any idea why Ahr never preceeds TF?

18. Fig 9. Lighter gray and darker gray are very hard to distinguish, choose another color legend.

Reviewer #2: Reviewer’s Report

Title: Deriving time-concordant event cascades from gene expression data: A case study for Drug-Induced Liver Injury (DILI)

Reviewer's report:

Authors of this manuscript study time concordance pathway cascades using the concept of first activation to generate hypotheses on potentially causal mechanisms following Bradford-Hill criterias. They use liver data from repeat-dose studies in rats to study time concordant gene expression-derived events preceding adverse histopathology, which serves as surrogate readout for Drug-Induced Liver Injury (DILI). While this manuscript is based on an interesting hypothesis, the manuscript could be improved.

Major Compulsory Revisions:

This study uses association analysis to discover the mechanisms of adverse outcome pathways. When a cause–effect relationship can be established based on first activation,

How do authors discriminate direct effects from indirect effects, especially with limited time points with gaps in time? For example, gene C is an unobserved confounder for the association between gene A and B but A is not a cause of B.

How do authors account for transient changes of the genes due to stochastic variations or biological fluctuations from unwanted external factors that may be random?

Some genes may appear too significant because of their role as network hub genes in the dataset that are involved in many biological processes. These hub genes are connected to many upstream regulators so their occurrence in cause-effect relationship could be general and unspecific unless integrated with prior knowledge. I would like to hear the author's response on how this could affect their inferences.

I) Methods:

Some of the important details of the methods are scattered in results and method sections. The methods section of this manuscript could be further improved by including details on methodology. Please refer to my comments below. I would strongly recommend a flowchart to improve readability.

Authors state in the abstract section: “from the TG-GATEs database which comprises measurements across eight timepoints, ranging from 3 hours to 4 weeks post-treatment”. How frequent are the time points? How does the gap in time points affect false predictions? Also, it is not clear how many biological replicates are used for different time points*dose.

While abstract mention TG-GATEs database which comprises measurements across eight timepoints, ranging from 3 hours to 4 weeks post-treatment, the results section states: “These adverse histopathology labels were next used to define 61 time-series associated, covering 38 compounds, as adverse (S2 164 Table).” It is not clear whether authors used 61 time series data which had similar time points.

“To evaluate which pathway or TF is dysregulated, we computed the differential activity in comparison to the time-, vehicle-, and experiment-matched control group using limma and identify significantly dysregulated gene sets with a False Discovery Rate (FDR) < 0.05.” What are the exact comparisons authors performed?

“For pathways and TFs, we defined first activation as the earliest time of measurement at which significant differential regulation was observed (FDR < 0.05) in each direction, while an additional logFC cut-off has been implemented for individual genes” For pathways and TFs, we defined first activation as the earliest time of measurement at which significant differential regulation was observed (FDR < 0.05) in each direction, while an additional logFC cut-off has been implemented for individual genes.” It is not clear how first activation of a pathway is defined? Do you consider a pathway to be activated even if one TF is expressed?

II) Results:

“These adverse histopathology labels were next used to define 61 time-series associated, covering 38 compounds, as adverse (S2 164 Table)”: What were the initial time-series and how did you define 61 time-series based on histopathology labels. If authors removed time points for downstream analysis, how does it bias the predictions?

“Fig 3: Enrichment of known events in DILI before adverse histopathology based on gene sets as well as individual gene members.” When you state gene sets, which comparisons did you perform? It is not clear from the writing.

Reviewer #3: This study has several strengths, including the time-resolved transcriptomics demonstations. Importantly, the authors should be commended for their comprehensive efforts. The manuscript is clear and easy to read.

A discussion of the results in relation to the more general issue of the TFs and Drug-Induced Liver Injury is lacking.

Reviewer #4: First, I’d like to thank Dr. Gallo, associate editor of PLOS Computational Biology, the opportunity to provide a peer-review for the work “Deriving time-concordant event cascades from gene expression data: A case study for Drug-Induced Liver Injury (DILI)” by Anika Liu and collaborators.

“Adverse outcome pathway, or AOP,” is a conceptual framework in toxicology that attempts to link a molecular initiating event with a defined adverse outcome. Liu and collaborators propose to identify novel potentially causal pairs of AOP-framework related key events (KE) and adverse outcomes (AO) by means of analysing temporal associations between these elements in an automated, data-driven way, with the aim of aiding in identifying novel AOPs in the future.

The authors propose to achieve this goal by using several metrics obtained from a time series dataset of liver gene expression deposited in the TG-GATEs database. These data, together with liver histopathologies, were obtained from the livers of rats that had been treated with a battery of xenobiotics. After normalising the microarray gene expression data, the authors used the first significantly differentially expressed gene as potential KE and capitalised on a previous exploitation of the same database that rendered scores of the histopathologies (as the AOs) to construct a model that captures the temporal dimension of the Bradford Hill criteria for causality [ref. 7 in the author’s manuscript] to prioritise associations with some support of causality. Being a pilar motivation of this work to capture causality in the associations detected, the authors limited their usable data to the perturbations, histopathology findings, and pathways that were relevant for the DILI pathology. This necessarily affected the capacity to generate pathway hypotheses that were purely data-driven, which is a central motivation of this work. However, despite the shortcomings that the authors nicely describe in a “Limitations of this study” section, they did a great job exploiting gene expression data to derive multiple metrics (significant enrichment, logFC and frequency/TPR) that combined maximise the ability to prioritise first event-outcome pairs from the database. This work is thus a valuable contribution to the global effort currently being undertaken to maximise the ability to exploit datasets deposited in countless databases, possibly serving not only as inspiration for novel ways of extracting important information from data, but also as a primer that can be further developed. Although I am not an expert on the toxicology domain, I believe that, given the linearity of the definition of AOP (Ankley et al., 2010 [ref. 5]; Leist et al, 2017 [ref. 4]), and the fact that it does not require full knowledge of the mechanism or pathway, the proposed method in this study is appropriate for its ambition of helping to establish new AOP hypotheses.

That said, I believe the readership would enjoy of a better flow if the descriptions of the datasets, assays and methods were more detailed. For example, although the authors make a substantial depiction of the extracted data on Figure 2C, which includes insight on the TG-GATEs assays study design, I still fail to find there or elsewhere in the manuscript important information: it does not include details like number of replicates performed per time-series, a description of how the serial or ‘’repeat-dose” treatment of animals was performed, what kinds of controls were used, how the rat livers were collected for histopathology and for gene expression measurement, strains of the animals, how much (or what fraction) of the dataset was excluded from the current analysis, etc… A supplementary figure detailing the TG-GATEs study design could be sufficiently effective in facilitating a better understanding of the methodology that was subsequently applied in this study.

Likewise, the authors should be more explicit in describing parameters and formulas, specifically of the hypergeometric distribution they used in their models. Additionally, they may want to describe in more detail some of the packages used. For example, on page 29 line 577, the authors refer the usage of the Limma package. The narrative could be more intuitive for the general audience if the authors add a brief description of methods applied.

The figures and tables in the main document and supplementary material are good and generally adequate, but the authors may want to add a Q-Q plot associated with figure 3 to show the calibration of their hypothesis test. The authors should also be more detailed and descriptive in some of the legends. I list below some cases that could be improved (e.g., adding a legend to File S2, and a better description of Table S2), but maybe the authors will find more opportunities of improvement at other instances.

The abundance of detail in presenting their findings of the case study overshadows the core contribution of a novel methodology to derive the time-concordant events in the first place. The manuscript would benefit greatly if those two aspects would be presented each with more focus. One way of achieving this could be to generalise the examples that the authors curated into schematic models that highlight the strengths of the model: where in the generalized examples is the model contributing that a non-data-driven, non-hypothesis-free method would not.

Finally, I believe that the authors will have a more engaged audience, enjoying of a better reading flow and understanding of the reasoning intended after the authors make a new round of careful editing, either themselves or with the help of a professional scientific editor.

I share below more specific comments and suggestions, by section in the main document. Note that the comments are not ordered by importance, but by approximate order of appearance in the manuscript. Where only a subsection tittle is shown, it means there are not further comments for that section or figure.

Abstract

1. Adverse Outcome Pathway is not generally known. I suggest the authors to link the expression “Adverse Outcome Pathway” with the term “toxicity” early in the abstract (optimally in first sentence), if they intend to help the reader immediately identify the domain of the work in the main abstract.

2. Page 2, line 33-34: When referring to “significance, frequency and log fold change (logFC)”, the authors may want to specify that “gene expression” is the object of the metrics they are referring to.

Introduction

3. Acronym inconsistencies:

Page 4, line 72: The AE acronym has not yet been introduced/defined in my version of the manuscript. The authors may want to define it here.

Page 4, line 76: The acronym for mode of action is MOA (as seen in Meek, 2014), therefore “mode of action” should include capitalisation of the three words, or no capitalisation at all. It could also be contained in brackets.

Page 4, line 87: Defining an acronym with another acronym may be too distracting for the reader. Maybe the authors would like to replace “KE relationships (KERs)” with “key event relationships (KERs)” for the sake of text fluidity/readability.

4. Definition inconsistencies:

Page 4, lines 77-80: The description of the criteria to evaluate plausibility may be confusing, in particular the explanations in parenthesis. It is not clear what “A” and “B” refer to. Later in the manuscript, “A” refers to a key event and “B” to an adverse outcome. This does not seem to be what the authors intend to convey here. For example, to describe “incidence concordance” the authors write “(The magnitude of event A is larger than that of event B)”. However, a quick check of the referenced work of Meek et al, 2014 shows incidence is explained with the question: “Is the occurrence of the end (adverse) effect less than that for the preceding key events?”. I fail to interpret the two explanations to mean the same.

It could be that I am completely missing the point of the authors, but for the sake of clarity and fluidity, the authors may want to put the general parameters/events described in the intro in context of the actual models and kind of data that is being presented throughout the manuscript (if needed be, with examples).

On identical note, in Figure 1 and across the manuscript, the authors may want to standardise the definitions of: “A”, “B”, “event”, “early event”, “later event”, “potential preceding event”, “potential later event”, “key event”, “anchoring event”, and “outcome”. It is not intuitive to use the word “event” for the “later event” and, if the authors believe it makes sense, they may want to distinguish “event” when it means “early event” from when it means “later event” by using e.g. the word “outcome” instead, across the manuscript, figures and legends, for the later (i.e., use “outcome” every time the word “event” is used in the sense of the “later event”). If, for some sensible reason this is not advisable, at least the word “event” should always be accompanied with the consistent (always the same) preceding word, e.g., “early” and “anchoring” (or “later”).

5. Figure 1 (Page 6, line 114):

Panel 1A: I understand that “A”,”B”,”C” and ”D” in panel 1A are instances of “A” in the model (“early event” or “potential preceding event”), as defined in the panel 1B, while “B”, in panel 1B (the model), is the “potential later event”. In Panel 1A, “A”,”B”,”C” and ”D” could be replaced by e.g., “I”,”II”,”III”, and “IV”. This way, “A” should always represent “preceding event” and “B” “later event or outcome” in the figure and across the manuscript.

Panel 1B) As already stated, in case the authors also believe it makes sense, they may want improve readability and consistency by defining “B” as “potential later outcome”.

Results and Discussion

Adverse histopathological findings and their temporal relation

6. Notation inaccuracy

Page 7, line 150: I believe the authors are referring to intervals, but the notation used is ambiguous. Using explicit intervals will help readership: “null” is within (0, 0.67] and “low” is (0.67, 1.34] ?

7. Typos

Page 7, line 138: The authors may want to use the plural of the word “hypothesis”, i.e., “hypotheses”

Page 8, line 163: The subject seems to be missing at the end of the excerpt: “labels were next used to define 61 time-series associated”.

Page 8, lines 165 and 166: If by “insult” the authors mean “adverse outcome”, it would be better to use the latter.

8. Figure 2 (Page 8, line 168)

Panel 2A: Colors are hard to distinguish: authors may want to increase label circles (in legend) and consider to change colors or, since there seems to be pairs of harder to distinguish colors, add a pattern to one within the pair.

Figure 2 legend:

Page 9, line 172: see comment for Page 7, line 150

Page 9, line 174: panel 2B does not show frequencies, but number of conditions.

Page 9, line 177: numbered reference missing after “Sutherland et al”.

Known pathways in DILI preceding adverse histopathology

9. Typos and omissions

Page 9, line 185: replace “(Fig 3).” With: “(Fig 3 and Table S3).”

Page 10, line 208: perhaps authors want to replace “processes except Growth Arrest And (…)” with: “processes except genes involved in Growth Arrest and (…)”

10. Figure 3 (Page 10, line 193)

- It would be useful to show how well the data is following the model: to that effect, the authors may want to add Q-Q plots to this figure.

Figure 3 legend:

Page 10, line 197: The LXR acronym is not defined anywhere in the manuscript.

11. Figure 4 (Page 12, line 229)

- This plot attempts to illustrate the strength of pathway perturbation. For the first panel, the authors may want to display the distribution of the “all” datapoint. This could be achieved by superimposing violin instead of box plots for this panel in particular. (The same extra information could be displayed in other figures anytime that the data is not sparse, usually, when the “all” datapoints are presented - Figure 3, Figure 6A and 6C).

12. Figure 5 (Page 13, line 239) and related Figure S3 (Supplementary file)

- It could be interesting to see how the data vary with varying FDR (actual precision-recall curves).

- Note that figure 5 is repeated, at least on my copy of the document: Page 13, line 239, line 242 and page 14 line 254 show the same image.

Figure S3, related with Figure 5 (Supplementary file)

- Although these plots expose a number of metrics, it is not intuitive what one can learn from them. The authors may want to expand a bit more.

Known TFs in DILI preceding adverse histopathology

13. Figure 6 (Page 16, line 297)

Figure 6 legend:

- It is not clear what “all” refers to (all the TFs in the genome?).

14. Add reference to a figure in text

Page 15, line 292: In the sentence “(…)as most frequent event indicating that the(…)”, the authors may want to add a reference to Figure 6B: “(…)as most frequent event (fig.6B) indicating that the(…)”

Data-driven prioritisation of cellular events taking place before adverse histopathology

15. File S2 (Supplementary file)

Page 16, line 311: File S2 - Is a supplemental spreadsheet, first referred on Page 10, line 202 - it would be useful if the authors added a legend describing the columns names, e.g., in a new tab.

16. Typos, clarification and nomenclature consistency

Page 17, line 323: In addition to the acronym, “UPR” should be described.

Page 17, line 320-331: The authors may want to contextualize the analysis described in the 4 sentences starting in “The most frequent(…)” and ending in “(…)the most strongly down-regulated TFs.” relative to their inclusion as known or plausible events in a more specific way, in order to cater for the readership that is not expert in DILI.

Page 17, line 329: Gene names: please make sure you use this journal’s gene names convention and use it consistently across the manuscript.

Figure 7

Identifying mechanistic hypotheses combining known TF functions and time concordance

17. TF activity metrics clarification

- The authors may want to explain how regulon activity is measured, or what is the output of the packages that extract this data (here or/and in the material and methods section, see my point 25.1).

18. Figure 8 (Page 19, line 359)

Figure 8 B: It’s unclear what the units for transcription factor activity are or mean.

19. Figure 9 (Page 21, line 408)

19.1 - It’s hard to distinguish what is diamond from circles because the gene names prevent visibility; perhaps the shapes size could be increased, for example, or/and the names could be non-superimposing the shapes, or the color scheme changed.

19.2 - The shadings of grey are nearly indistinguishable for the important metrics of “Only at the same time” or “Ordered or at the same time”, authors may want to use a different color scheme

Time-concordant events reflecting disease progression

20. Figure 10 (Page 23, line 444)

Figure 10 legend:

Page 23, line 448 - typo in word “correlation”

Limitations of this study

Conclusions

21. Transforming specific pathway examples into a generalised model schematics depicting the strengths (and possibly shortcomings) of the method:

In addition to detailing the pathways themselves, it would be more useful to the readership interested in the methodology to use generalised terms that would explain the capabilities of this model. One way the authors could achieved this is by drawing a schematic illustrating how the author’s method was able to confirm an example of known events in DILI, and contrasting it with other generalised example(s) where the model was able to fill in the gaps in knowledge, at different levels.

22. Page 27, line 524: Link to Shiny app is missing at this location

Methods

Open TG-GATES data processing

Definition of adverse histopathology

23. Expand on methodology details

Page 28, line 563: More details should be given. Please be explicit about the parameters and equation used in the hypergeometric test. Also, please provide intuition for the choice of hypothesis test (i.e. why the hypergeometric distribution).

Page 28, line 559: The authors may want better define DILIst, DILIrank and explain table S2 further.

24. Typos

Page 27, line 539: Typo: Replace “To characterize the extend…” with “To characterize the extent…”

Page 27, line 542-3: And replace: ”(…)and then summarised across all biological replicates by mean as an aggregate(…)” with: “(…)and then averaged across all biological replicates as an aggregate(…)”

Page 27, line 550: see comment 6. (Page 7, line 150)

Pathway and TF activity inference

25. Expand on methodology details

25.1 - Page 28, line 567-8: The authors may want to give a more granular description of the TF activity output by the package(s) mentioned. How does TF activity data look like? It could also be useful to include a short explanation of the choices made.

25.2 - Page 28, line 575-7: The authors may also want to exactly describe which methods were used in addition to the packages that assisted on the implementation of those methods.

25.3 - A thorough description of the TC-GATEs dataset must also be shared in order to effectively understand the analysis. As suggested earlier, this could include a figure.

25.4 - In addition to a schematic figure describing the assays, it would be most useful to include more schematics of the author’s pipeline.

Temporal concordance of events

26. Page 30, line 598: The authors may want to share the parameters and equation used in the hypergeometric test.

**Have the authors made all data and (if applicable) computational code underlying the findings in their manuscript fully available?**

Reviewer #1: None

Reviewer #2: Yes

Reviewer #3: Yes

Reviewer #4: None

PLOS authors have the option to publish the peer review history of their article (what does this mean?). If published, this will include your full peer review and any attached files.

Reviewer #1: No

Reviewer #2: No

Reviewer #3: No

Reviewer #4: No
---

## [Decision Letter · Decision Letter 1]

26 Apr 2022

Dear Ms. Liu,

We are pleased to inform you that your manuscript 'Deriving time-concordant event cascades from gene expression data: A case study for Drug-Induced Liver Injury (DILI)' has been provisionally accepted for publication in PLOS Computational Biology.

Best regards,

James Gallo

Associate Editor

PLOS Computational Biology

Mark Alber

Deputy Editor

PLOS Computational Biology

Reviewer's Responses to Questions

**Comments to the Authors:**

Reviewer #1: My comments are addressed in the revised manuscript.

Reviewer #3: None

Reviewer #4: The authors have satisfactorily answered to my concerns.

**Have the authors made all data and (if applicable) computational code underlying the findings in their manuscript fully available?**

Reviewer #1: None

Reviewer #3: Yes

Reviewer #4: None

PLOS authors have the option to publish the peer review history of their article (what does this mean?). If published, this will include your full peer review and any attached files.

Reviewer #1: No

Reviewer #3: No

Reviewer #4: No

---

## [Editor Report · Acceptance letter]

30 May 2022

PCOMPBIOL-D-21-02220R1 

Deriving time-concordant event cascades from gene expression data: A case study for Drug-Induced Liver Injury (DILI)

Dear Dr Liu,

I am pleased to inform you that your manuscript has been formally accepted for publication in PLOS Computational Biology. Your manuscript is now with our production department and you will be notified of the publication date in due course.

With kind regards,

Zsanett Szabo
